# Feasibility and acceptability of gamified cycling exercise for residents in a long-term care home: A qualitative study

Lillian Hung[1,2]*, Zhiqi Shen[3], Joey Cheung[4], Michelle Lam[2], Jamie Lam[2], Tiffany Wu[2], Rikki Wu[4], Arwen Fong[2], Michelle Xiao[2], Riea Singh[2], Yang Qiu[5], Lily Wong[2], Yong Zhao[2]

1 School of Nursing, University of British Columbia, Vancouver, British Columbia, Canada, 2 Innovation in Dementia and Aging Lab, University of British Columbia, Vancouver, British Columbia, Canada, 3 College of Computing and Data Science, Nanyang Technological University, Singapore, 4 Villa Cathay Care Home, Vancouver, British Columbia, Canada, 5 Joint NTU-UBC Research Centre of Excellence in Active Living for the Elderly (LILY), Nanyang Technological University, Singapore

* lillian.hung@ubc.ca

## Abstract

Gamification can motivate older adults to exercise by transforming physical activities into enjoyable experiences. Incorporating gaming elements in cycling exercises can foster a sense of interest and achievement, potentially improving health outcomes. This study investigated the acceptability and feasibility of motivating residents living in a long-term care (LTC) home with a gamified cycling exercise. Fourteen residents completed a 4-week gamified cycling exercise twice a week. Safety during exercise was addressed by assessing heart rate and observation. With an interpretive description approach, we conducted observations and interviews with residents and family members and focus groups with staff and leadership. The thematic analysis identifies three themes representing the feasibility and acceptability of gamified cycling exercise among LTC residents: ease of use and accessibility, physical and mental health benefits, fun engagement and community building. Future research should explore dementia-friendly design, culture-related game content, family orientation and engagement, group exercise and organization support. This study showed the promising acceptability and feasibility of gamified cycling exercise in an LTC home. Successful implementation relies on tailoring interventions to meet residents' specific needs and preferences while acquiring rapport with interdisciplinary staff in the care home.

## Introduction

Aging is often associated with a decrease in physical activity and the development of metabolic disorders and other chronic diseases [1]. The Centers for Disease Control recommends that most adults get 150 minutes of moderate-intensity physical activity

**Data availability statement:** All relevant data are within the paper and its Supporting Information files.

**Funding:** This work was supported by the Canada Research Chair in Senior Care awarded to LH (Grant Number: GR021222). The funders had no role in the study design, data collection and analysis, decision to publish, or preparation of the manuscript. (Http://www.chairs-chaires.gc.ca/chairholders-titulaires/profile-eng.aspx?profiledId=5178). There was no additional external funding received for this study.

**Competing interests:** The authors have declared that no competing interests exist.

a week, but in real life, only 15% of adults over 65 years old get a sufficient amount of exercise [2]. People who are inactive are 50% more likely to have chronic diseases (e.g., diabetes, hypertension, obesity, and depression) than those who are physically active [3]. However, even if older adults are familiar with the potential benefits of exercising, they are still not motivated to do so. This situation is even more serious in long-term care (LTC) homes, as residents might have complex health situations and are feeling more isolated.

Game-based technology, also known as gamification, may provide a promising solution to motivate exercise among older adults [4,5]. Gamified exercises are often named exergames. Exergames are digital games that use motion-sensing technology to track body movements and provide real-time feedback, blending physical exertion with gaming for fun and improved health [6]. Gamification has become increasingly popular across health and exercise [7]. Despite the widespread adoption, gamification in healthcare has primarily targeted children to engage them in physical exercise for fitness [8], as they are deemed more digital. However, with the limited studies available, the data shows that the use of gamification could have positive results for older adults across a wide range of health areas as well [9–11].

A scoping review of health games urged future studies to broaden their focus to include more diverse demographics, particularly older adults [12]. Emerging evidence suggests that older adults can benefit greatly from gamification, especially in the cognitive and psychological, as well as physical health [9,13,14]. Gamified interventions have been shown to enhance engagement and motivation in physical activity [15–17] while also promoting positive emotions. Beyond the health and psychosocial benefits, gaming technologies provide additional advantages, such as offering older adults a fresh and enjoyable activity while remaining affordable for both individuals and care providers [5].

Despite the positive outcomes, factors such as usability, safety, and the specific needs of LTC residents are often overlooked. Some studies focus solely on attendance rates as a measure of feasibility without providing detailed insights [18]. This highlights the need for empirical studies in LTC to rigorously test the feasibility of exergames. Additionally, while most studies assess balance and mobility improvements or reduced fall risks [13], 53.3% of older adults in LTC homes in British Columbia, Canada, rely on wheelchairs, with some cases exceeding 70% [19]. These residents cannot be assessed using traditional balance and mobility measures. This gap necessitates research into exergame implementation and its impact on frailer older adults with dementia, a population that remains significantly underexplored in existing research [13,20,21]. Furthermore, current studies in gamified interventions often focus solely on residents or only on staff [22,23], which may overlook the broader dynamics at play in LTC settings. It is also important to understand the opinions of different partners, such as family members and staff. Their insights can provide valuable context regarding the implementation and effectiveness of these interventions [24,25]. Exploring these views offers a deeper understanding of the complexity involved in such programs, as staff frequently operate the interventions, and family members can provide valuable input based on their intimate knowledge

of residents' needs and expectations [25]. Incorporating perspectives from residents, staff, and family allows for a more comprehensive approach to delivering effective gamified interventions in LTC.

Accessing the acceptability and feasibility of the cycling game is essential to understand both its perceived value and its practical integration in LTC settings. In this study, acceptability refers to the extent to which residents, family members and staff perceive the gamified cycling exercise as agreeable, engaging, and valuable. Feasibility is defined as the extent to which the intervention can be practically implemented within the LTC environment, considering factors such as staff workload, resident suitability, integration into existing routines, and technological infrastructure.This dual focus is essential for identifying potential barriers and facilitators to broader implementation across similar care settings. This study investigates the acceptability and feasibility of implementing a gamified cycling exercise program tailored specifically for older adults living in LTC homes from all key stakeholders—residents, family members, LTC staff and leadership. This comprehensive approach reflects the collaborative nature of care in LTC environments, where residents' uptake and staff facilitation are closely intertwined. While findings may inform future scale-up, this study is primarily concerned with identifying facilitators and barriers within the current care environment.

Research Questions:

1. What are older adults' experiences of using a digital game to motivate cycling exercise in a Canadian LTC home?

2. What are the perceptions of families, interdisciplinary staff, and leadership regarding using the game to motivate cycling exercise in the Canadian LTC home?

## Methods

### Design

We applied the Interpretative Descriptive (ID) approach [26]. Using the ID approach to study the feasibility and acceptability of a cycling game exercise in LTC is appropriate because ID allows for a deep exploration of experiences and behaviors in real-world clinical settings. Qualitative data in the ID applied approach provides us with rich information explaining the reasons (the "why") and the approaches (the "how"). ID is particularly useful for generating practical implementation knowledge that can directly inform clinical practice for wider adoption in LTC settings. We used focus groups with staff and practice leaders in the LTC home. We also conducted observations and conversational interviews with residents and families to gather diverse perspectives on the program's acceptability and feasibility. Focus groups fostered group synergy and reflective collaboration [27], while conversational interviews provided a more intimate setting for vulnerable populations to share nuanced insights, complementing the broader perspectives gathered in the group discussions [28]. Over a four-week period, 14 residents from an LTC in Western Canada participated in a structured program involving gamified cycling exercises conducted twice a week. Our paper follows the Consolidated Criteria for Reporting Qualitative Research (COREQ) by Tong et al. [29], with detailed reporting provided in Supporting Information S1 Table.

### Study settings and the cycling game

This study was conducted in a non-profit LTC, housing around 170 multicultural residents, 70% of whom have dementia and other comorbidities, with a staff-to-resident ratio of 1:10. The cycling program was implemented in the care home's recreation center using an Active Passive Trainer (APT) connected to a TV that displayed a gamified program with high-definition animations, vibrant colors, and music to engage residents. A sensor on the APT tracked residents' pedalling, which controlled the pace of an animated fox chasing rabbits on screen. Researchers briefed participants, explaining that the cycling helped the fox catch rabbits. During the intervention, we monitored residents' performance and safety, using field notes to track heart rate with an oximeter, signs of sweating, fatigue, ability to talk, and asking if they felt tired. See Fig. 1.

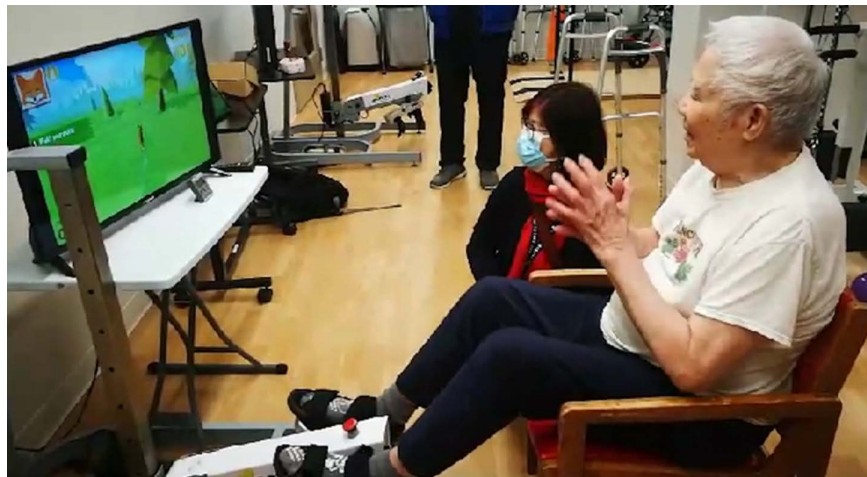

**Fig 1. The Gamified Exercise Equipment and Programme.**

## Sampling and recruitment

We utilized a convenience sampling method [30] to recruit participants through posters in the care home. Emails were sent to staff and leaders as well as families to invite participation. The inclusion criteria for participants included: (1) aged over 60; (2) residing in LTC homes; (3) currently participating in routine exercise without a gamified program; and (4) able to communicate in simple English and willing to participate. The exclusion criteria for participants included (1) severe cardiovascular conditions, (2) advanced respiratory diseases, (3) actively undergoing treatment and regularly visiting the hospital, and (4) not meeting the above inclusion criteria. For family members, the inclusion criterion was the ability to understand English. Staff and leadership participants had to be full-time or part-time employees at the LTC home in Vancouver, British Columbia, Canada. The recruitment period of this study spanned from June 12, 2024, to August 15, 2024. To respect the diverse composition of frontline staff and families from racialized communities, we did not require proficiency in English for participation and placed no additional exclusion criteria as long as the inclusion criteria were met. A demonstration video summarizing the project was shared with staff and family participants prior to the focus groups and interviews (https://youtu.be/be4WBJwC4gM).

## Focus group and interview questions

The questions for the interview for residents were:

1. What do you like or dislike about the game?

2. What are the good and bad of this game?

These two questions focused on acceptability, reflecting residents' emotional and cognitive responses to the intervention.

The questions for the semi-structured focus group for LTC staff and interviews with families were:

1. What are the benefits and concerns of this cycling game?

2. What suggestions do you have for making improvements to this game?

3. What are the barriers and facilitators in implementing gamified exercise across LTC homes? (only for leadership participants)

The first two questions addressed both acceptability and feasibility, depending on participants' perspectives and emphasis in their responses. The last question was designed to explore feasibility, particularly organizational and operational factors, to scale the intervention.

## Data collection

This project was led by the first author, LH, who is an experienced gerontological nurse in LTC. Interviews were conducted with both residents and family members; however, these were conducted separately due to differing schedules in this study setting. Each family member was interviewed individually at their preferred time points, while residents were interviewed directly after their final exercise session. The aim was to gather in-depth reflections on their overall experience with the intervention rather than session-specific feedback. We conducted 11 focus groups with LTC staff and leadership between June and August 2024. Each focus group consisted of two to seven participants, based on the participants' availability and workload. There was no repetition in focus group participation; each staff member attended only one discussion session. Focus groups were organized by role type for scheduling and comfort reasons. Specifically, allied health professionals participated in one group, managers in another, and nurses and care aides were grouped together in several sessions due to their different working sites. This structure also helped avoid the potential influence of hierarchical dynamics in the focus group discussions. Researcher trainees YZ, ML, TW, and JL conducted these focus groups at the nursing stations of the care home. Each session was audio-recorded and transcribed verbatim. Observation field notes were also recorded during the group sessions to document non-verbal cues and contextual factors that might impact the conversation. Our team included diverse trainees: JL, RS, ML, MX, TW, YZ and AF. YZ is an Asian male health leadership and policy graduate. JL is a female undergraduate student in Cognitive Science. AF is an Asian female undergraduate student in Integrated Sciences. ML, MX, and TW are Asian female bachelor's students in nursing. RS is an Asian female undergraduate student in Speech Sciences. All data collection activities by trainees were under the supervision of LH, the principal investigator of the study, who is a female Asian professor in nursing.

## Data analysis and theoretical framework

Our iterative data analysis was guided by the consolidated framework for implementation research (CFIR) [31]. CFIR informed us in analyzing how individual and contextual factors impacted the effectiveness of gamified exercise programs in LTC settings. Our study utilized Braun and Clarke's [32] six-step reflexive thematic analysis approach. Step 1: Trainees transcribed the data, and all research team members read independently. With this, the team oriented itself to the data. Step 2: YZ and JL performed initial analysis and generated the initial codes with the transcribed data. This study used the NVivo version 14.0 program to store all the transcripts, quotations, and codes. Based on the CFIR framework and research questions, the team developed a system for deductive and inductive coding. To enhance methodological rigor, a subset of transcripts was independently coded by two researchers (YZ and JL) to establish inter-coder reliability. Discrepancies were resolved through discussion and consensus. Step 3: Based on the codes developed, YZ, ML, TW, and JL searched for common themes. Step 4: YZ presented and discussed the preliminary themes with the research team. Step 5: Based on the team discussion, we refined the themes. Step 6: The trainees wrote the initial draft and revised it through several iterations. The supervisor LH provided guidance to the student authors during the data analysis and writing process. Descriptive statistics, including frequencies and percentages, were used to analyze demographic variables such as age range, gender distribution, ethnicity, dementia status, and staff roles.

## Patient and public involvement

One family partner, LW, was involved in organizing focus groups and interviews, as well as in data analysis. She had previously worked with the first author on several research projects (e.g., [33,34]). LW brought valuable insights from her lived experience as a caregiver for LTC residents, helping to ensure that the research addressed the real-world needs of residents and caregivers. Additionally, she led staff huddles and informal meetings to share research progress in the LTC home.

### Ethical considerations

This research study was approved by the Office of Research Ethics at the University of British Columbia (H24-00010). Written consent was gathered from all participants through a consent form discussing the details of the research, outlining its purpose, potential benefits, risks involved, and the right to withdraw from the study. Documentation of consent forms is securely maintained as part of the study records. To maintain confidentiality, pseudonyms were employed to protect the anonymity of participants' identities. The resident in Fig 1 and their family have given written informed consent (as outlined in PLOS consent form) to publish these case details. To protect participant privacy, full transcripts and anonymized data are not available in a public repository. However, coding frameworks and selected de-identified excerpts are included in this manuscript to support transparency.

### Rigor

This study implements several measures to ensure rigor in the output research, including the active involvement of family partners for credibility. Our team incorporated regular reflective discussions into weekly research meetings and kept reflexivity notes to critically evaluate our assumptions. Team members with different disciplinary backgrounds brought diverse perspectives to data interpretation, supporting triangulation of analytic insights. Our team consists of multiple researchers collaborating on coding, analysis, and interpretations. A subset of transcripts was double-coded to enhance dependability, and discrepancies were reconciled through consensus-building discussions. Detailed research documentation, description of data collection, and analysis processes were documented in an audit trail. These practices, including triangulation, inter-coder checks, and transparent documentation, were established to continually promote accountability and transparency in our research.

## Results

### Participants

The study included 59 participants, including 14 residents, 11 family members, and 34 staff, whose ethnicity is all Asian (see Table 1; demographic data are available in the Supporting Information, S2 Dataset). Staff are from various professional backgrounds, with nurses (29.4%) and care aides (29.4%) representing the largest groups. Recreational team (20.6%) and rehabilitation team (5.9%) also participated. Notably, the leadership group comprised two nurses, one rehabilitation therapist, one recreational therapist, and one manager. The gender distribution was predominantly female (82.4%). Staff's ages ranged from 20 to 60 years, with the largest age group being 50–60 years (35.3%), followed by 30–40 years (29.4%), 40–50 years (23.5%), and 20–30 years (11.8%). This diverse demographic provided a comprehensive perspective on the implementation of the gamified cycling program.

The gender distribution of residents was predominantly female (78.6%). Participants' ages ranged from 68 to 96 years, with the largest age group being older than 90 (64.3%), followed by 80–89 years (21.4%). The majority of residents are living with moderate dementia (57.1%).

All participants reported positive impact brought by this cycling game and expressed their anticipation on the next phase of this program. One family member Leilani mentioned, *"It seems like my dad really enjoyed it, and I'm excited to see what's next. Will he be involved in the next phase as well?"* This research created three themes: ease of use and accessibility, physical and mental health benefits, fun engagement and community building. See Table 2.

### Theme 1: Ease of use and accessibility

**Ease of use.** All residents agreed that the game was easy to play. Family members and staff expressed that the cycling game was user-friendly for most residents, regardless of their cognitive abilities or physical limitations. Scarlett, a care aide, observed,

**Table 1. Demographic Characteristics of Participants (n = 59).**

| Residents | | Families | | Staff | |
|---|---|---|---|---|---|
| Gender | | | | | |
| Male | 3 (21.4%) | Male | 1 (9.1%) | Male | 6 (17.7%) |
| Female | 11 (78.6%) | Female | 10 (90.9%) | Female | 28 (82.4%) |
| Age group | | | | | |
| 65-79 years | 2 (14.3%) | 40-49 years | 3 (27.3%) | 20-30 years | 4 (11.8%) |
| 80-89 years | 3 (21.4%) | 60-69 years | 7 (63.6%) | 30-40 years | 10 (29.4%) |
| 90-100 years | 9 (64.3%) | 70-79 years | 1 (9.1%) | 40-50 years | 8 (23.5%) |
| | | | | 50-60 years | 12 (35.3%) |
| Ethnicity | | | | | |
| Asian | 14 (100%) | Asian | 11 (100%) | Asian | 34 (100%) |
| Dementia | | | | Roles | |
| None | 2 (14.3%) | | | Caring team | |
| Mild | 3 (21.4%) | | | Nurse | 10 (29.4%) |
| Moderate | 8 (57.1%) | | | Care aide | 10 (29.4%) |
| Severe | 1 (7.1%) | | | Recreation team | 7 (20.6%) |
| | | | | Rehabilitation team | 2 (5.9%) |
| | | | | Leadership | 5 (14.7%) |
| Total | 14 | | 11 | | 34 |

**Table 2. Codes and Themes.**

| Themes | Subthemes | Codes |
|---|---|---|
| Ease of use and accessibility | Ease of use | easy to set up, easy to understand, easy to use, friendly for disabilities, auditory cues for visual impairment, culture-related game content, language subtitle, dementia-friendly design, pedalling position, screen brightness, game showcase integration, vibration for visual impairment |
| | Accessibility | avoiding responsibility, cognitive impairment exclusion, cost-effectiveness, generational understanding gap, health concerns, leadership permission, no weekend coverage, cycling is better than walking, integrating into routine, monitoring for safety, implementing in assisted living |
| | Resource constraints | APT equipment shortage, no workload added, staffing shortage, design reduces staffing, group exercise reduces staffing, independence reduces staffing, staff workload |
| Physical and mental health benefits | Physical health benefits | build up muscles, desire to be independent, feeling energetic, increased physical activity, promoting circulation, reducing pain, resident education, tired after exercise, no change noticed |
| | Mental health benefits | cognitive health, emotional recall, forget exercise, learning, maintaining cognitive capacity, memory training, mental health, enjoyment, social interaction, mutual benefits |
| Fun engagement and community building | Fun engagement | expectation for exercise, fun experience, game elements motivation, gift motivation, monotonous without game, reluctant to exercise, residents' interest, alternative entertainment choice, award motivation, choice available, encouragement, food rewards, fresh experience, refuse routine, multi-level design, tired yet persistent, visual engagement, vocal prompts |
| | Community building | changes on staff perception, collaboration between staff and family, family anticipation, family concerns on safety, family orientation, family support, group exercise, competition design, competitive content, role model of residents, initial invitation, interdisciplinary collaboration, ongoing invitation, staff support, staff training, online training, supervise required |

*"I do see the residents actually doing it, and they don't look confused at all, so I think it's a good sign that the physical activity has been successfully achieved."*

*"I have dry eyes, but I find it easy to keep them open while watching the fox because the colors are just right for me,"* said Stella, a resident.

*"There's a limit, though; sometimes a certain cognitive level is required for residents to fully participate. However, even those in wheelchairs can still join in,"* noted Lune, a nurse.

Staff remarked that the game setup seemed straightforward, as it was the same for all residents. The only challenge they faced was ensuring the proper seating adjustments for each individual.

*"It seems to be easy because you just have to have one set up, right, for all of the residents, so it's the same for everybody,"* commented Emery, a family member.

Staff and family members also offered suggestions for improving the game's usability and acceptability, such as incorporating culturally relevant content (e.g., elements tied to residents' long-term memories), language subtitles, and dementia-friendly design features. These included simpler designs, avoiding rapid motion, using larger screens and targets, offering choices without overwhelming the participants, providing two different color set options for residents with visual impairments, and providing auditory and vibration cues. They also recommended creating a comfortable environment, with reduced brightness behind the TV and well-designed chairs, to ensure proper seating and distance.

### Accessibility

Staff and family members identified several facilitators for the accessibility of the cycling game, such as its integration into current exercise routines, available monitoring and supervision, and its suitability for residents in wheelchairs. Ruby, a family member, commented, *"They don't have to use as much strength since they're just seated and cycling. Walking is more tiring because it requires weight-bearing. So, I think this is a great intervention."*

Leilani, another family member, shared, *"I know that in the past, he did this type of cycling during physio, so I was confident he could do it again. It's great to see him back at it! Plus, since he likes video games, combining the two has been very beneficial for him."*

However, some staff highlighted a few barriers to the game's feasibility, such as exclusion criteria for residents with cognitive impairments, concerns about safety, staffing shortages, leadership approval, and accountability in case of accidents.

Zoey, a care aide, explained, *"Some residents aren't very interested or have more complicated conditions, so it might not be suitable for their floor."*

Oliver, a recreation staff member, remarked, *"I think the project coordinator or the department head will try to integrate it into their program, and that will be their responsibility. In the future, a couple of months down the line, they can start spreading it out to suitable residents. However, I believe it should start with the rehab team of professionals to lead the initial efforts."*

Lily, in leadership, noted, *"The goal of exercise can differ depending on whether it's part of an activity program or a rehab program. By incorporating it into an activity program, I think more residents can have access to it."*

Families and staff also expressed initial hesitation about involving certain residents due to health concerns, such as the risk of heart attacks. However, after observing the heart rate monitoring and supervision during the exercise, they felt reassured about the project's sustainability and scalability. One family member remarked, *"Wow, so you're measuring them, that's awesome! I'm curious though, how do you know if they're tired or not? Then I found there was something (oximeter) hooked up to them. That's great!"*

### Resource constraints

Resource constraints have been identified as a significant barrier to the successful implementation of this project in LTC homes. While most of the care team did not feel that the program increased their workload, both leadership and other teams highlighted staffing and equipment shortages as major challenges.

Lily, from the leadership team, explained, *"The main barrier is still staffing. Not many residents are in a PT program because the goals of exercise differ. However, if it's part of an activity program instead of rehab, more residents could participate. But since this is a one-on-one game, it requires more dedicated staff because only one resident can play at a time, even though others can watch – they still have to line up."*

She also pointed out the issue of accessibility: *"Accessibility is the second challenge – how can we get more residents involved at the same time? We've tried other games where three or four can play together, but this one is just one at a time. I remember some residents asking, 'When is it my turn?' So, improving accessibility and creating smoother arrangements will definitely be a challenge."*

Suggestions to address these constraints were provided by families, staff, and leadership. These included increasing residents' independence, enhancing the game design, and introducing group exercise options.

Gianna, another member of leadership, remarked, *"If only one or two residents participate, it won't be cost-effective. For example, under the management's funding criteria, one recreation team covers 32 to 36 residents. One recreational staff member is responsible for residents on two floors, so we expect at least 5 to 8 residents to participate at the same time."*

### Theme 2: Physical and mental health benefits

**Physical health benefits.** Given the limited mobility of many residents, families, staff, and residents themselves highlighted the ease of improving physical and mental health through participation in the cycling game. Residents and their families appreciated the opportunity to engage in physical activity, with several participants reporting that they felt more energetic, experienced less stiffness in their knees, and had looser legs after playing. Many participants saw the game as a way to strengthen their muscles, improving their sense of mobility. Natalie, a family member of a wheelchair-bound resident, remarked that the game was *"a really great tool for them to keep their feet moving,"* noting that without the ability to walk, residents risk losing muscle mass.

Ruby, another family member, shared her thoughts: *"It just kinda gets the blood going, and for those who don't have good mobility in their legs, it provides some form of exercise. Especially when exercise is hard, and they can't really stand up, it helps them build some strength this way."*

Families also appreciated how the gamification of cycling helped residents improve their coordination. Natalie added, *"Having a supervised program where they can coordinate their feet and realize, 'Oh, it's a game, I'm chasing something,' gets the brain going. It makes the coordination feel purposeful, like they're not just pedaling for no reason—they're chasing something."*

### Mental health benefits

The cycling game offered varying cognitive benefits depending on the residents' cognitive levels. While some residents could not recall the game, others remembered it fondly, reporting that recalling the experience made them laugh. Staff noted that the cycling game provided residents with opportunities to learn new things, helping to maintain their attention and cognitive functions. Everly, a family member, shared, *"It can help her learn to focus since she's completely lost track of things. This can help her regain that focus."*

Many family members and staff also noted that the cycling game increased social interaction opportunities, positively impacting residents' mental health. Luna, a nurse, commented, *"Most people here feel quite lonely and are seeking attention. If someone sits beside them and offers encouragement, it makes them really happy."* Similarly, Joshua, a family member, observed, *"Right now, no one really interacts with each other. I'm hesitant to talk to the neighboring resident because I don't know them or their family. But an activity like this could increase interactions between families and residents, improving the social aspect. It would bring people together instead of everyone living on the same floor without ever speaking."*

Lainey, a recreation staff member, highlighted the combined benefits of exercise and other activities, stating, *"The body and mind are connected, so doing exercise is definitely good for other parts of the body too. When I do art with them, I start with some warm-up exercises or body movements. This helps them feel more energetic, uplifts their mood, and makes them feel warmer. Then they're ready to create art."* This insight suggests the potential for integrating exercise with other recreational activities.

Physical and mental health benefits make the game more acceptable because it delivers meaningful outcomes, such as improved mobility, cognitive focus, and social interaction. From a feasibility perspective, the cognitive stimulation provided by the game, alongside its potential to enhance social interaction, meant that it could serve multiple purposes—physical exercise and mental health improvement. This dual purpose makes the intervention more efficient and easier to justify within a resource-constrained environment. Additionally, the social aspect increased residents' willingness to participate, reducing the need for one-on-one supervision and making it more feasible to integrate into group activities.

### Theme 3: Fun engagement and community building

**Fun engagement.** Residents expressed pride in the time they spent participating in the cycling game. Maya, one of the residents, praised her ability to cycle for 10 minutes each session and expressed a desire to cycle for the full duration every time.

Natalie, a family member, appreciated the new form of engagement, commenting, *"There's a certain amount of social interaction because they don't do it alone. And they have a team, so they feel more important or well-cared for."*

Ruby, another family member, shared her perspective on how the game improved her mother's enthusiasm for exercise:

*"Before this game (program) began, when my mother was first living in this care home, she would go up to the training room and cycle while facing the wall. I think she feels bored cycling and then loses enthusiasm after a while. I think this is a very interesting program. There's a colorful screen, and it's a bit of a game for them to play. As you know, older adults can be eager to win at games. If they know that they've caught a rabbit, then they'll be really happy. From the videos I saw, I could see that the residents were really happy cycling. I hope that this sparks their interest in exercising."*

Staff members also observed that residents were more engaged in this activity compared to other recreational events, often looking forward to the next session. They appreciated how the game added a motivating element, making it feel less like a chore and more like an enjoyable, interactive experience. Luna, a nurse, noted, *"Sometimes there are a lot of excuses like, 'I'm tired' or 'I've been sitting for so long.' So having something to watch or an aim to increase their score can help increase their concentration. And it'll increase their interest and motivation to cycle or to cycle longer."*

Mia, a leadership member, also praised the engagement created by the game, saying, *"A lot of times residents say, after a short while, they want to leave the program, but you're able to keep their attention. So that's a good thing, and also just seeing them smile—that's something I'm really proud of."*

Some family members even commented on the residents' anticipation for the game. Natalie mentioned, *"They're wondering when the game will come back because it's been a positive experience for my parents. I even told them it was just an experiment and might not return, and they seemed a bit disappointed."*

The residents' engagement has been attributed to several factors, such as the game's elements, visual effects, vocal prompts, gift incentives, and avoidance of monotony, which kept them invested in the game even when they felt tired. Leilani, a family member, remarked, *"It's a great way to get the residents to do physical activity with a goal in mind. They're exercising while also being engaged, like playing a game at the same time."* Ana, a care aide, added, *"When they go back to their room and see the toy (gift), it'll remind them of the game."*

Participants also offered suggestions for further enhancing resident engagement, such as more encouragement, food rewards, and improvements to the game design (e.g., more options, various levels tailored to residents, and adding

rewards for motivation). Oliver, a staff member, suggested, *"If a game has levels—like one, two, and three—people naturally want to challenge themselves, even older adults. If you keep it the same, it gets boring, but adding a little variation makes a big difference."*

The fun engagement makes the cycling game highly acceptable by ensuring residents are motivated, entertained, and socially connected. These aspects make participation feel less like a task and more like an enjoyable activity, which increases the willingness of both residents and families to embrace the intervention.

### Community building

The game not only sparked social interaction and excitement among residents, who eagerly anticipated each session, but it also fostered a fun atmosphere that encouraged collaboration between staff and families. This collaboration strengthened support for the residents and contributed to building a sense of community. Residents expressed a need for staff to be beside them, providing encouragement and support through auditory cues and assistance with setting up the APT equipment. While some staff noted that a few residents preferred solitude in their rooms, many more expressed a desire to cycle together with friends. Both staff and family members recognized the happiness derived from group exercise and offered suggestions for improving the program.

For example, one family member, Joshua, suggested, *"Competing with their neighbour to see who can score the most points will also boost their motivation."* Recreational staff member Oliver added, *"You add the player into the game, and then they start interacting, trying different things. The game is divided into different sections and windows, where they can track each other. You can keep everything in the same setting but make it more interesting to engage them."*

Staff emphasized that familiarity is essential for building close relationships with residents. Oliver explained, *"When you first start working with a senior, you're not really familiar with them yet – you don't know their limitations or strengths. But after a few times, as you build a relationship, you get to know them better. Everyone has different backgrounds and needs, and once you understand that, it becomes easier to keep things moving smoothly, right?"*

Many family members expressed their anticipation of exercise engagement and their passion for motivating residents to seize more exercise opportunities and leverage this meaningful program. Joshua remarked, *"Often, when family members visit residents, they're unsure how to spend the time. A cycling game that helps residents exercise for 10-20 minutes would be a great way to make their visit more meaningful."* He also recalled the challenges of collaboration during the COVID-19 pandemic, suggesting, *"The staff didn't want families to start nitpicking, so there was resistance to the idea (of collaboration). But if everyone shares the same goal, there's no need to officially name a family representative – they'll naturally take on that role and help nearby residents exercise."*

Both families and staff acknowledged the necessity of supervision during exercise for safety concerns. Penelope, a leadership staff member, noted, *"When it's in passive mode, it just keeps spinning non-stop, which is a concern. So, when residents use it, we always have close supervision. All it takes is one slip, and they could seriously fracture something. That's why I find that part a bit worrisome."* Sophie, a family member, added, "I feel like if it wasn't supervised, my grandma might get a little too competitive." Another family member, Natalie, felt reassured knowing that safety can be guaranteed: *"I don't really see a downside (to this program) as long as they're being supervised. It's just the physical limitations we have to watch out for."*

To scale this program effectively, family and staff emphasized the importance of training and orientation. Oliver suggested, *"To make it more successful, I think creating a short video or an online course would help. Everyone could then meet in a room for a brief training session."*

The collaboration among all partners is crucial for the success of community building, which in turn enhances the program's feasibility. The focus on staff familiarity with residents facilitates smoother operations and creates a more supportive environment for both staff and participants. Additionally, the emphasis on training and orientation highlights the practicality of scaling the program, as clear guidance can lead to its effective implementation across various care settings.

The attention to supervision and safety protocols further ensures that the program remains viable and safe, particularly in LTC settings. All these factors contribute to the program's sustainability and scalability, making it a promising intervention for enhancing the well-being of residents.

The following summary integrates the key themes in relation to the study's two research questions, demonstrating how the data supports our conclusions on acceptability and feasibility. Together, the three key themes—ease of use and accessibility, physical and mental health benefits, and fun engagement and community building—offer a comprehensive understanding of the acceptability and feasibility of implementing a gamified cycling game in LTC. Feasibility and acceptability are not mutually exclusive in this study; rather, they are interconnected aspects that together inform the successful implementation of a gamified cycling program in LTC settings. Each theme includes two to three subthemes that address both the user experience and practical realities of the LTC context. Themes related to ease of use, fun engagement, and perceived physical and mental health benefits address acceptability from the perspective of residents, aligning with the first research question on older adults' experiences. However, acceptability also emerged in the perspectives of staff and family, particularly regarding ease of use and perceived health improvement. In contrast, subthemes related to accessibility, resource constraints, and community building reflect broader systematic and contextual factors influencing feasibility, speaking to the second research question regarding staff, family, and leadership perceptions. Notably, some themes, like ease of use and perceived health benefits, reflect an overlap between acceptability and feasibility; the positive responses from participants suggest that greater willingness to adopt the intervention can reduce resistance, increase engagement, and support long-term sustainability. Although we began with conceptually distinct definitions—acceptability as perceptions of value and engagement, and feasibility as practical implementation within LTC contexts—our analysis revealed that these constructs frequently intersect in practice. To maintain analytical clarity, we interpreted overlapping themes through the lens of stakeholder role and context, allowing us to capture how acceptability and feasibility manifest differently yet complementarily across levels of care. These themes offer a comprehensive view across individual, interpersonal, and organizational levels, guiding future implementation strategies for the successful adoption of gamified intervention in LTC environments.

## Discussion

Our results contribute to advancing the field of exergames by addressing critical gaps in the existing literature, particularly regarding a frail older adult population living with dementia in LTC. Previous research predominantly focused on younger cohorts with mild or no cognitive challenges [20,21]. In contrast, our study stands out by including 64.3% of participants aged over 90 and 64.2% with moderate to severe dementia, demonstrating the feasibility of implementing exergames for an older and more cognitively impaired population. Our findings point out the unique challenges and potential benefits for this group.

From the analysis, the three themes that emerged were ease of use and accessibility, physical and mental health benefits, fun engagement, and community building. It was demonstrated that the participants found the gamified cycling exercise easy and in support of both their physical and mental well-being. The intervention also fostered engagements and community building promoting social interactions among residents and staff.

Similar to previous research, residents, family members, and staff in our study highlighted the way in which the game provided participants with a form of entertainment during cycling, thereby increasing their motivation to cycle [35,36]. Focus group participants also emphasized how the game elements play a key role in maintaining motivation, and the game's potential for more personalization options and aspects of competition for further engagement and acceptability. This demonstrates that our current game aligns with prior research on factors that increase the appeal of exercise to older adults, but the feedback from interviewees provides an avenue for further exploration in this game's development [37,38].

Research on exergames for older adults demonstrates positive impacts on physical and mental health, though findings vary in strength. For instance, Karssemeijer et al. [39] found that exergames significantly increased exercise adherence

and effectively reduced frailty levels in older adults with dementia compared to traditional physical exercise groups. Similarly, Cai et al. [20] highlighted improvements in cognitive and physical capacities, such as balance ability and gait, in older adults with MCI and dementia, although these assessments may not fully apply to our resident group, many of whom are wheelchair-bound. Additionally, Koivisto and Malik [21] reported positive outcomes across multiple domains, including visual attention, diabetes control, and increased physical activity, though most effects were only weakly supported. The integration of social elements and dynamically adjustable difficulty was emphasized by Karaosmanoglu et al. [40], as these factors can enhance the engagement and safety of exergames. Moreover, Brookman et al. [37] observed improvements in functional fitness, depression, self-efficacy, and social interaction in an international cycling competition intervention, underlining the multimodal benefits of such programs for the well-being of older adults. Through our interviews, it was found that family members and staff emphasized the physical and mental benefits of cycling, which contributed to their positive perceptions of the gamified cycling exercise. Though family members raised concerns about residents' and families' knowledge of the importance of exercise, many interviewees demonstrated an existing understanding of the physical benefits of cycling. However, in contrast to the family members' acknowledgements of the role of the gamified exercise in providing mental stimulation, residents largely focused on the physical benefits, such as muscle building and relief from musculoskeletal. This discrepancy is noteworthy, as research has indicated that exercise can provide cognitive benefits, such as maintaining one's attention and processing speed [41]. The limited mentions of cognitive benefits of exercise indicate a need to examine whether residents are aware of these perks. Exploration of resident's understanding of physical activity outcomes is important, as prior research has identified the lack of education to be barriers to exercise [42,43]. As such, understanding the residents' level of knowledge and educating those who are unaware of the cognitive stimulation that exercise provides may increase engagement and participation in exergames, particularly in individuals who value cognitive wellbeing. This may also suggest the importance of understanding what older adults view to be mentally stimulating, and whether this perception influences their willingness to engage in exercise as a form of cognitive stimulation.

While our thematic analysis was guided primarily by the CFIR framework, which focuses on multilevel implementation processes, the Technology Acceptance Model (TAM) [44] and the Unified Theory of Acceptance and Use of Technology (UTAUT) [45] are widely used frameworks for assessing how users accept and adopt technology. TAM focuses on two primary factors: *perceived usefulness* and *ease of use*, which together predict the likelihood of technology acceptance [44]. UTAUT expands on TAM by incorporating four key determinants of intention and usage: *performance expectancy* (expected physical and mental health benefits), *effort expectancy* (ease of use), *social influence* (support from family and staff), and *facilitating conditions* (accessibility and organizational support) [45]. Although these frameworks did not guide our coding or theme development, they provided a useful comparative lens. These theories provide valuable insights into the factors influencing the acceptance of exergames among older adults. Several constructs from TAM and UTAUT are reflected in the themes derived from our CFIR-informed analysis. For example, the theme of "ease of use and accessibility" overlaps with UTAUT's effort expectancy and facilitating conditions; "physical and mental health benefits" aligns with performance expectancy; and "fun engagement and community building" relate to social influence and perceived usefulness. These factors align with the three main themes of our study, which were derived from comprehensive data collection from multiple perspectives, ensuring a well-rounded understanding of the factors influencing the acceptability of the gamified cycling exercise in LTC settings. In line with previous findings, such as Catherine Park [46], which reported good acceptability of exergames among older adults with dementia and mild cognitive impairment (MCI), our study echoed similar outcomes. A scoping review of exergame usability and acceptability for older adults also supports the promising potential of such interventions besides physical benefits like postural balance and muscle training [47]. The design of the gamified cycling game in our study, which incorporated user-friendly technology and simplified game mechanics, enabled participation from a wide range of residents facing various physical and cognitive challenges, such as mobility limitations, arthritis, reduced muscle strength, and cognitive decline. The intuitive setup allowed residents to engage with the exercise without heavy reliance on staff guidance, and the game's accessibility—suitable for those seated in wheelchairs or with

limited mobility—made it inclusive for those who may not be able to engage in traditional exercise programs. This adaptability reduced the intimidation often associated with new technologies or exercise programs, particularly for older residents, and encouraged higher participation rates. When interpreted through the lens of TAM and UTAUT, the cycling game demonstrated strong ease of use and overall acceptability, driven by its ability to accommodate diverse resident needs and encourage active participation. These findings highlight the feasibility of using gamified exercise interventions in LTC settings, offering an inclusive solution for residents with varying physical and cognitive limitations.

Social interaction and community support are particularly important in LTC settings, where residents often face social isolation, which can negatively impact their well-being [48]. The gamified cycling exercise intervention played a significant role in fostering a sense of community and engagement among residents. The interactive nature of the program provided multiple opportunities for social interaction, both by allowing residents to observe and support each other during the exercise and by facilitating communication with staff members. Through participation in the cycling game, residents were able to form new social connections, contributing to a stronger sense of belonging within the LTC community.

Family members play a crucial role in alleviating staffing shortages in LTC settings by actively engaging in resident care. Research shows that family involvement enhances care quality, fosters community, and improves residents' well-being [49]. Our findings reveal that family members are passionate about motivating residents to seize exercise opportunities, highlighting their commitment to enhancing experiences. Appointing project coordinators can facilitate communication and collaboration between families and staff, ensuring that family members are informed and can effectively contribute to care plans. Project coordinators also play a vital role in coordinating interdisciplinary care teams, such as rehabilitation and recreational staff, which enhances program implementation and aligns team goals [50]. Promoting family engagement through structured programs creates a supportive environment that mitigates staffing challenges [51]. This approach strengthens relationships among staff, families, and residents, contributing to a sustainable gamified exercise model. Additionally, staff emphasized the importance of familiarity with residents, further illustrating families' vital role in fostering a supportive atmosphere.

## Strengths and limitations

This study highlights several strengths, including the novel approach of incorporating gamified exercise into LTC, which fosters both physical and cognitive engagement among residents. A key strength of this study is the inclusion of a significantly older population (64.3% over 90 years old) and a high proportion of residents with moderate to severe dementia (64.2%), addressing gaps in previous research that largely focused on younger, less cognitively impaired individuals. The inclusion of various partners – residents, families, and staff – allowed for a comprehensive understanding of the program's acceptability and feasibility from multiple perspectives. The study also demonstrates how gamification can enhance motivation and engagement, as residents reported enjoying the interactive nature of the game. The collaborative nature of the study, with input from leadership and caregivers, further strengthens its practical applicability to real-world settings.

Despite its strengths, the study has some limitations. The relatively small sample size and the demographic composition, primarily of Chinese individuals from a single LTC facility, may restrict the generalizability of the findings. The demographic homogeneity of our sample was not intentional selection but a natural consequence of recruiting at the available site. While this reflects the local institutional and cultural context, it was not a deliberate choice to limit diversity. As such, the findings may not capture the perspectives or experiences of residents, families, and staff in other regions, cultural contexts, or care models. For instance, attitudes toward technology use, physical activity, or family involvement in care may vary across cultural or socioeconomic groups. In addition, organizational policies, staffing structures and recreational programming can differ across LTC facilities, which may influence how interventions like gamified cycling are received and implemented. To further examine the acceptability and feasibility of this cycling game, future research involving multiple long term care facilities, a broader diversity in ethnicities and cultures, and a larger number of individuals is needed. Additionally, this study primarily focused on qualitative data. Incorporating quantitative measures such as heart rate variability,

loneliness and happiness scales, and quality of life assessments in future studies could provide a more holistic understanding of the program's effectiveness. Future research could examine the cost-benefit analysis and long-term impacts of gamified cycling exercises in improving residents' physical and mental health outcomes, helping LTC facilities make informed decisions about the program's feasibility and suitability for boarder implementation.

**Practice implications**

Based on the insights shared by residents, family, and multidisciplinary staff working in LTC homes, there are some lessons learned/ practical tips - "PEDALING" for all partners when implementing gamified exercises in LTC:

- **Personalized and Inclusive Design**: Customize the game to accommodate residents with diverse cognitive and physical abilities (e.g., simplified controls, color sets, subtitles, culture-related content). This ensures inclusivity and ease of use for a broad range of participants.

- **Engagement Through Group Play:** Modify the game to support group activities and friendly competition. This fosters social interaction, community building, and encourages more residents to participate, boosting overall engagement.

- **Digital Tools for Scalability**: Create online tutorials and training modules to facilitate implementation across different care settings. This will help streamline staff onboarding and enable the game to scale more effectively to a wider and larger study population.

- **Adaptability into Routine Care**: Integrate the game into existing care routines by aligning it with both rehabilitation and recreational programs. This improves feasibility without significantly increasing staff workloads.

- **Leadership and Family Involvement:** Encourage family participation to motivate residents and empower family members to engage meaningfully with their loved ones. Leadership support is crucial in coordinating the program's integration and expansion.

- **Improving Motivation with Game Elements:** Incorporate rewards, varying levels of difficulty, and visual/auditory cues to keep residents engaged. These elements prevent monotony and encourage sustained participation, making the game enjoyable and effective.

- **Necessary Supervision and Safety Protocols:** Ensure close supervision during gameplay, especially for residents with mobility challenges. Safety measures like heart rate monitoring and protocols to prevent overexertion are vital to maintaining program viability and acceptability.

- **Goal-Oriented Physical and Cognitive Health Monitoring:** Use the game to track residents' physical and cognitive improvements (e.g., muscle strength, attention span). Regular feedback loops between staff, residents, and families will optimize individual care plans.

## Conclusion

In summary, our study demonstrates that a gamified cycling exercise program is both acceptable and feasible for residents in LTC homes. The intervention was well-received by participants, staff, and families, with notable benefits in physical health, mental engagement, and social interaction. Successful implementation of such programs requires careful attention to the unique needs and preferences of residents, ensuring accessibility for those with varying levels of mobility and cognitive function. Furthermore, fostering a collaborative environment between staff, families, and residents enhances the program's effectiveness, as building rapport is crucial for maintaining engagement and providing the necessary support. The findings suggest that gamified exercise interventions have the potential to be a sustainable, scalable approach to improving the well-being of older adults in LTC settings, while also addressing physical, cognitive, and social health

challenges. Further research is encouraged to explore long-term outcomes and strategies for broadening participation across diverse resident populations.

## Supporting information

**S1 Table. COREQ checklist.**
(DOCX)

**S2 Dataset. Dataset used for analysis in this study.**
(XLSX)

## Acknowledgments

We would like to express our gratitude to Jesse Hui and Sharon To from Villa Cathay Care Home for their invaluable assistance in facilitating the residents' exercise sessions, interviews, and focus groups. We also extend our heartfelt thanks to all the staff and family members who contributed their time and insights, making this project possible through their active participation and support. Special thanks to Kara for her contributions in transcribing the interviews, which greatly aided our analysis. Your efforts and commitment are deeply appreciated.

## Author contributions

**Conceptualization:** Lillian Hung.

**Data curation:** Zhiqi Shen, Joey Cheung, Michelle Lam, Jamie Lam, Tiffany Wu, Rikki Wu, Arwen Fong, Michelle Xiao, Riea Singh, Yang Qiu, Lily Wong, Yong Zhao.

**Formal analysis:** Lillian Hung, Zhiqi Shen, Joey Cheung, Michelle Lam, Jamie Lam, Tiffany Wu, Rikki Wu, Arwen Fong, Michelle Xiao, Riea Singh, Yang Qiu, Lily Wong, Yong Zhao.

**Funding acquisition:** Lillian Hung.

**Project administration:** Yong Zhao.

**Supervision:** Lillian Hung.

**Writing – original draft:** Lillian Hung, Michelle Lam, Jamie Lam, Tiffany Wu, Rikki Wu, Arwen Fong, Michelle Xiao, Riea Singh, Yang Qiu, Lily Wong, Yong Zhao.

**Writing – review & editing:** Lillian Hung, Zhiqi Shen, Joey Cheung.

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
