## [Decision Letter · Decision Letter 0]

19 Feb 2025

Dear Dr. Hung,

Thank you for submitting your manuscript to PLOS ONE. After careful consideration, we feel that it has merit but does not fully meet PLOS ONE’s publication criteria as it currently stands. Therefore, we invite you to submit a revised version of the manuscript that addresses the points raised during the review process.

We look forward to receiving your revised manuscript.

Kind regards,

Amir Karimi, PhD

Academic Editor

PLOS ONE

**Journal Requirements:**

Please ensure that your manuscript meets PLOS ONE's style requirements, including those for file naming. The PLOS ONE style templates can be found at https://journals.plos.org/plosone/s/file?id=wjVg/PLOSOne_formatting_sample_main_body.pdf and https://journals.plos.org/plosone/s/file?id=ba62/PLOSOne_formatting_sample_title_authors_affiliations.pdf 2. Thank you for stating in your Funding Statement: This work was supported by the Canada Research Chair in Senior Care awarded to LH (Grant Number: GR021222). The funders had no role in the study design, data collection and analysis, decision to publish, or preparation of the manuscript. (https://www.chairs-chaires.gc.ca/chairholders-titulaires/profile-eng.aspx?profileId=5178)Please provide an amended statement that declares *all* the funding or sources of support (whether external or internal to your organization) received during this study, as detailed online in our guide for authors at http://journals.plos.org/plosone/s/submit-now.  Please also include the statement “There was no additional external funding received for this study.” in your updated Funding Statement. Please include your amended Funding Statement within your cover letter. We will change the online submission form on your behalf. 3. Please amend either the abstract on the online submission form (via Edit Submission) or the abstract in the manuscript so that they are identical. 4. We note that Figure 1 includes an image of a participant in the study. As per the PLOS ONE policy (http://journals.plos.org/plosone/s/submission-guidelines#loc-human-subjects-research) on papers that include identifying, or potentially identifying, information, the individual(s) or parent(s)/guardian(s) must be informed of the terms of the PLOS open-access (CC-BY) license and provide specific permission for publication of these details under the terms of this license. Please download the Consent Form for Publication in a PLOS Journal (http://journals.plos.org/plosone/s/file?id=8ce6/plos-consent-form-english.pdf). The signed consent form should not be submitted with the manuscript, but should be securely filed in the individual's case notes. Please amend the methods section and ethics statement of the manuscript to explicitly state that the patient/participant has provided consent for publication: “The individual in this manuscript has given written informed consent (as outlined in PLOS consent form) to publish these case details”.  If you are unable to obtain consent from the subject of the photograph, you will need to remove the figure and any other textual identifying information or case descriptions for this individual.

**Additional Editor Comments:**

Hello, dear authors

The respected reviewers of the journal have given a positive opinion on your article and we hope to accept your article with a few corrections. However, the editor would like to use it for the next round of review to complete the information in your article.

Please adapt the article completely to the journal format.

There are numerous errors in grammar and English writing, including capitalization problems.

Avoid capitalizing paragraphs.

Especially, format the tables in a consistent manner.

We await your response and wish you all the best.

Reviewers' comments:

Reviewer's Responses to Questions

**Comments to the Author**

1. Is the manuscript technically sound, and do the data support the conclusions?

Reviewer #1: Yes

Reviewer #2: Yes

2. Has the statistical analysis been performed appropriately and rigorously?

Reviewer #1: Yes

Reviewer #2: Yes

3. Have the authors made all data underlying the findings in their manuscript fully available?

Reviewer #1: Yes

Reviewer #2: Yes

4. Is the manuscript presented in an intelligible fashion and written in standard English?

Reviewer #1: Yes

Reviewer #2: Yes

**Reviewer #1: ** The authors discussed the manuscript in a clear manner and in detail. Ample references were used to support their findings as well as in the literature review section. The research done was crucial to understand how exergames could benefit older community.

**Reviewer #2:**  I have no competing interests.

The work was done perfectly. Also it is have few minor notes to adding: like statistical program and other statistical ways used in showing research results.

Secondly, the study should be built-in wider and large studying group.

**Do you want your identity to be public for this peer review?** For information about this choice, including consent withdrawal, please see our Privacy Policy

Reviewer #1: No

Reviewer #2: **Yes: ** Yahya Ali Abdulkareem Abodea

---

## [Author Response · Author response to Decision Letter 1]

5 Mar 2025

Journal Requirements:

Feedback: Thanks for your suggestions. We make sure that our manuscript meets PLOS ONE’s style requirements.

This work was supported by the Canada Research Chair in Senior Care awarded to LH (Grant Number: GR021222). The funders had no role in the study design, data collection and analysis, decision to publish, or preparation of the manuscript. (https://www.chairs-chaires.gc.ca/chairholders-titulaires/profile-eng.aspx?profileId=5178)

Feedback: Thanks for your suggestions. We have amended funding statement as follows and add “There was no additional external funding received for this study”:

Funding statement

This work was supported by the Canada Research Chair in Senior Care awarded to LH (Grant Number: GR021222). The funders had no role in the study design, data collection and analysis, decision to publish, or preparation of the manuscript. (http://www.chairs-chaires.gc.ca/chairholders-titulaires/profile-eng.aspx?profiledld=5178). There was no additional external funding received for this study.

Feedback: Thanks for your comments. We have reviewed both the abstract in the manuscript and the one in the online submission form and conform that they are identical. There are no amendments necessary.

4. We note that Figure 1 includes an image of a participant in the study.

Feedback: We securely filed the signed consent form in the individual’s case notes and revised the manuscript accordingly (page 12, line 186-187).

Feedback: We review our reference list and ensure that it is complete and correct.

Review Comments to the Author

Reviewer #1: The authors discussed the manuscript in a clear manner and in detail. Ample references were used to support their findings as well as in the literature review section. The research done was crucial to understand how exergames could benefit older community.

Feedback: Thanks for your positive comments. Your comments are highly encouraging for us to explore the implementation of gamified exercise further.

Reviewer #2: I have no competing interests.

The work was done perfectly. Also it is have few minor notes to adding: like statistical program and other statistical ways used in showing research results.

Secondly, the study should be built-in wider and large studying group.

Feedback: Thanks for your comments. We report participant numbers by age range, gender distribution, ethnicity, dementia status, and staff roles using descriptive statistics (numbers and percentages). This information has been added to the Methods section (Page 11, Lines 170-172). Beyond that, no additional data is available for further statistical analyses.

This study represents the first phase of implementing gamified exercise in long-term care homes. We assessed its acceptability and feasibility in the current study and plan to explore its effectiveness in future research while expanding participation across diverse resident populations. We have addressed these limitations in Page 33, Lines 552-560, and outlined our future plans in the Implications (page 34, line 574) and Conclusion sections (page 36, line 604-605.

---

## [Decision Letter · Decision Letter 1]

10 Apr 2025

Dear Dr. Hung,

Thank you for submitting your manuscript to PLOS ONE. After careful consideration, we feel that it has merit but does not fully meet PLOS ONE’s publication criteria as it currently stands. Therefore, we invite you to submit a revised version of the manuscript that addresses the points raised during the review process.

We look forward to receiving your revised manuscript.

Kind regards,

Amir Karimi

Academic Editor

PLOS ONE

Journal Requirements:

Reviewers' comments:

Reviewer's Responses to Questions

**Comments to the Author**

Reviewer #3: All comments have been addressed

Reviewer #4: (No Response)

2. Is the manuscript technically sound, and do the data support the conclusions?

Reviewer #3: (No Response)

Reviewer #4: Yes

3. Has the statistical analysis been performed appropriately and rigorously?

Reviewer #3: (No Response)

Reviewer #4: Yes

4. Have the authors made all data underlying the findings in their manuscript fully available?

Reviewer #3: (No Response)

Reviewer #4: Yes

5. Is the manuscript presented in an intelligible fashion and written in standard English?

Reviewer #3: (No Response)

Reviewer #4: Yes

Reviewer #3: The authors went through the comments from both reviewers and made the necessary changes in the manuscript. Also considered future application on a wider scale.

Reviewer #4: Introduction

-Please include the definitions and specific context of feasibility and acceptability in your introduction. Acceptance of the motivated cycling exercise by the LTC home, patient, family member, LTC staff, or all the above? Feasibility of the motivated cycling experience to be implemented at a larger scale, or in a specific context, and how is feasibility being determined?

-Unclear how the research questions directly relate to acceptance and feasibility of the cycling game.

Methods

-Please clarify which interview and focus group discussion questions address acceptance and which address feasibility. Factors of acceptance and feasibility can be mentioned to indicate if there is a focused approach.

-Please confirm if the interviews were conducted with family members or patients or both. “After completing each exercise session (twice weekly for four weeks), we conducted 20-30- 143 minute individual conversational interviews with residents and their families.” Were multiple interviews conducted with each resident, one after each exercise session?

-Please clarify if the staff focus group discussions were mixed (multiple roles/ranks in one group)? Please also clarify if there was repetition in focus group discussion participation, such as one staff member attending multiple discussions.

-Did the researchers take into account the multiple levels of rank in one discussion group? Was the possibility of bias accounted for in the focus group discussions? For example a subordinate being uncomfortable to openly criticize the intervention in the presents of leadership which brought in the intervention.

Results

-Accessibility is the first the author is mentioning feasibility from the LTC home’s perspective, goes back to earlier question of clarifying the acceptance and feasibility parameters. Is the focus of the study to assess acceptability and feasibility by the patients or the care providers, or both? Staff discussing the interventions affect on the patient and how to bring the intervention to the patient is still patient focused, but I do not currently see how lines 257-260 provide evidence of feasibility/acceptability

-Which themes and sub themes speak to acceptability by the patient/LTC home versus which speak to feasibility for the patient/LTC? Why were the themes included? Please link the themes to research your questions relating to acceptability and feasibility.

- Consider breaking up the results in feasibility and acceptability sections/subsections to ensure clear findings

Discussion

-lines 513 mention as assessed through TAM, was the study using the TAM framework throughout? If so the TAM should be mentioned in the introduction, with its definitions and factors of acceptability

-discussion highlights the perceived benefits of the cycling game thoroughly, but please also include detailed discussion of the other factors relating to acceptability and feasibility.

Conclusion

-No comments

-The paper is surely contributing to implementation practices in elder care with innovate care technology. There is a clear contribution to knowledge in the areas of elderly care and the use of gamified technology in healthcare. The data was well collected, but the data is approached and presented without clear criteria on the definitions or factors of acceptability and feasibility.

**Do you want your identity to be public for this peer review?** For information about this choice, including consent withdrawal, please see our Privacy Policy

Reviewer #3: **Yes: ** Marwa Said

Reviewer #4: No

---

## [Author Response · Author response to Decision Letter 2]

24 Apr 2025

Reviewers' comments:

6. Review Comments to the Author

Reviewer #3: The authors went through the comments from both reviewers and made the necessary changes in the manuscript. Also considered future application on a wider scale.

Response: Thanks for your comments.

Reviewer #4: Introduction

-Please include the definitions and specific context of feasibility and acceptability in your introduction. Acceptance of the motivated cycling exercise by the LTC home, patient, family member, LTC staff, or all the above? Feasibility of the motivated cycling experience to be implemented at a larger scale, or in a specific context, and how is feasibility being determined?

Response: Thanks for your comments. We have now defined acceptability as the extent to which residents, family members, and staff perceive the cycling game intervention as agreeable, engaging, and valuable. Feasibility is defined as the extent to which the intervention can be practically delivered within the long-term care settings, considering staffing, scheduling, and infrastructure. The study explores acceptability and feasibility from the perspectives of all key stakeholders—residents, family members, LTC staff and leadership. This comprehensive approach reflects the collaborative nature of care in LTC environments, where residents’ uptake and staff facilitation are closely intertwined. In terms of scope, our assessment of feasibility focuses on the implementation of the cycling game within the specific context of the participating LTC homes. We examine factors such as workflow integration, staff capacity, and resident suitability. While findings may inform future scale-up, this pilot study is primarily concerned with identifying facilitators and barriers within the current care environment. We added relevant content in Page 5 (lines 78-91).

-Unclear how the research questions directly relate to acceptance and feasibility of the cycling game.

Response: Thanks for your comments. We clarify that both research questions are designed to explore key dimensions of acceptability and feasibility (Page 5, lines 78-85). The first question examines older adults’ experience, which provides insight into acceptability from the resident perspective—whether they find the cycling game engaging, appropriate, and enjoyable. The second question captures the perspectives of family members, staff and leadership, offering critical insight into both the acceptability (e.g. perceived value) and feasibility (e.g., workflow integration, staff capacity, and operational fit) of implementing the intervention in LTC. These perspectives together inform the potential for sustained use and broader implementation.

Methods

-Please clarify which interview and focus group discussion questions address acceptance and which address feasibility. Factors of acceptance and feasibility can be mentioned to indicate if there is a focused approach.

Response: Thanks for your comments. We revised the manuscript to explicitly map these questions to the constructs of acceptability and feasibility (Page 8, lines 146-147, 154-156).

-Please confirm if the interviews were conducted with family members or patients or both. “After completing each exercise session (twice weekly for four weeks), we conducted 20-30- 143 minute individual conversational interviews with residents and their families.” Were multiple interviews conducted with each resident, one after each exercise session?

Response: Thanks for your comments. We confirm that interviews were conducted with both residents and family members; however, these were conducted separately due to differing schedules in this study setting. Each family member was interviewed individually at their preferred time points, while residents were interviewed after their final exercise session. The aim was to gather in-depth reflections on their overall experience with the intervention rather than session-specific feedback. We have clarified this in the revised manuscript to avoid confusion and to better reflect our data collection approach (Page 8, lines 159-163).

-Please clarify if the staff focus group discussions were mixed (multiple roles/ranks in one group)? Please also clarify if there was repetition in focus group discussion participation, such as one staff member attending multiple discussions.

Response: Thanks for your comments. We confirm that there was no repetition in focus group participation; each staff member attended only one discussion session. Focus groups were organized by role type for scheduling and comfort reasons. Specifically, allied health professionals participated in one group, managers in another, and nurses and care aides were grouped together in several sessions due to their different working sites. We have clarified this point in the revised manuscript (Page 9, lines 165-171).

-Did the researchers take into account the multiple levels of rank in one discussion group? Was the possibility of bias accounted for in the focus group discussions? For example a subordinate being uncomfortable to openly criticize the intervention in the presents of leadership which brought in the intervention.

Response: Thanks for raising this important point. We acknowledge the potential influence of hierarchical dynamics in focus group discussions. Our focus group structure also helps avoid this influence. Additionally, the topic of discussion, feasibility and acceptability, was not directly related to job performance or sensitive organizational matters. We have clarified this in the revised manuscript (Page 9, lines 165-171).

Results

-Accessibility is the first the author is mentioning feasibility from the LTC home’s perspective, goes back to earlier question of clarifying the acceptance and feasibility parameters. Is the focus of the study to assess acceptability and feasibility by the patients or the care providers, or both? Staff discussing the interventions affect on the patient and how to bring the intervention to the patient is still patient focused, but I do not currently see how lines 257-260 provide evidence of feasibility/acceptability

Response: Thanks for your comments. The aim of our study was to explore acceptability and feasibility from the perspectives of all three groups: residents, family members, and care providers. The staff quote on lines 257-260 illustrates feasibility from the organizational and staffing perspective. Oliver’s comments specifically reflect considerations around leadership responsibility, structured implementation through the rehabilitation team, and the phased approach to scale the intervention across appropriate residents. While the content centers on patients, the shared insight is fundamentally about how feasible it is for staff and the organization to deliver and expand the intervention.

-Which themes and subthemes speak to acceptability by the patient/LTC home versus which speak to feasibility for the patient/LTC? Why were the themes included? Please link the themes to research your questions relating to acceptability and feasibility.

Response: Thanks for your helpful comments. We have added a paragraph at the end of the results section to explicitly link the identified themes and subthemes to our research questions, as well as the concepts of acceptability and feasibility (Page 24, lines 459-477). This addition clarifies how each theme aligns with the experiences and perspectives of residents, families, staff and leadership, and enhances the overall coherence of the findings.

- Consider breaking up the results in feasibility and acceptability sections/subsections to ensure clear findings

Response: Thanks for your suggestion. While we recognize the value in distinguishing feasibility and acceptability in some research designs, we intentionally chose an integrated thematic approach in this study. Our rationale is that feasibility and acceptability are not mutually exclusive; rather, they are interconnected aspects that together inform the successful implementation of a gamified cycling program in LTC settings. For example, themes such as “Ease of use and Accessibility” speak directly to both the practicality of setup (feasibility) and user satisfaction (acceptability). Separating them could risk oversimplifying these nuanced intersections. We have clarified this rationale in the manuscript and ensured that each theme is linked to our research questions to highlight how they collectively reflect feasibility and acceptability (Page 24, Lines 459-477).

Discussion

-lines 513 mention as assessed through TAM, was the study using the TAM framework throughout? If so the TAM should be mentioned in the introduction, with its definitions and factors of acceptability

Response: Thanks for your comments. We did not use TAM framework to guide our thematic analysis; rather, our themes were developed inductively from the data. However, TAM is an important framework related to understanding acceptability, so we referenced it in the discussion section to help contextualize our findings and support interpretation.

-discussion highlights the perceived benefits of the cycling game thoroughly, but please also include detailed discussion of the other factors relating to acceptability and feasibility.

Response: Thanks for your comments. Beyond perceived health benefits, we also discussed ease of use as a key factor in acceptability and suggested design improvements to enhance user engagement and comfort (Page 27, lines 530-558). In terms of feasibility, we highlighted fun engagement and community building as important facilitators (Page 29, Lines 559-566), along with the role of families in addressing staffing challenges and supporting implementation (Page 29, lines 567-580). These factors collectively provide a more nuanced understanding of what supports or hinders the adoption and sustainability of gamified cycling in LTC settings.

Conclusion

-No comments

-The paper is surely contributing to implementation practices in elder care with innovate care technology. There is a clear contribution to knowledge in the areas of elderly care and the use of gamified technology in healthcare. The data was well collected, but the data is approached and presented without clear criteria on the definitions or factors of acceptability and feasibility.

Response: Thanks for your comments. We integrated these definitions and related factors into the manuscript. (Page 5, lines 78-91)

---

## [Decision Letter · Decision Letter 2]

7 Jun 2025

Thank you for submitting your manuscript to PLOS ONE. After careful consideration, we feel that it has merit but does not fully meet PLOS ONE’s publication criteria as it currently stands. Therefore, we invite you to submit a revised version of the manuscript that addresses the points raised during the review process.

We look forward to receiving your revised manuscript.

Kind regards,

Amir Karimi, PhD

Academic Editor

PLOS ONE

Journal Requirements:

Reviewers' comments:

Reviewer's Responses to Questions

**Comments to the Author**

Reviewer #5: (No Response)

Reviewer #6: (No Response)

2. Is the manuscript technically sound, and do the data support the conclusions?

Reviewer #5: Partly

Reviewer #6: Yes

3. Has the statistical analysis been performed appropriately and rigorously?

Reviewer #5: Yes

Reviewer #6: Yes

4. Have the authors made all data underlying the findings in their manuscript fully available?

Reviewer #5: Yes

Reviewer #6: Yes

5. Is the manuscript presented in an intelligible fashion and written in standard English?

Reviewer #5: Yes

Reviewer #6: Yes

Reviewer #5: Hello author, thank you for providing this manuscript for review. I have shared several comments, primarily highlighting its strengths. However, I've also noted some areas where improvements could be made.

1. Technical Soundness and Data-to-Conclusion Alignment

The study is presented as a qualitative inquiry examining the acceptability and feasibility of a gamified cycling exercise program in a long-term care (LTC) setting. The authors carefully define essential concepts. For instance, in lines 78–91, "acceptability" is defined as how engaging and valuable the residents, family members, and staff find the intervention, whereas “feasibility" refers to practical aspects like staffing and scheduling. This dual focus is well justified in the context of the study.

The methodological approach is thoroughly described (e.g., an Interpretative Descriptive approach noted in lines 97–105 and the iterative coding process using NVivo discussed in lines 181–190). The methods for observations, individual interviews, and focus groups are suitably explained, and incorporating perspectives from multiple stakeholders (residents, families, staff, and leadership) enhances the conclusions drawn.

Quotations from participants (e.g., lines 245–254 and lines 378–391) effectively illustrate the emergent themes (such as ease of use, physical/mental health benefits, and enjoyment/community engagement) rooted in the data. Along with summary tables (Tables 1 and 2 referenced around lines 231 and 242), these data points provide credible support for the manuscript’s conclusions.

The conclusions regarding feasibility and acceptability (lines 638–650) are well grounded in the study findings and existing literature (e.g., lines 500–529).

Despite the overall strong technical design, it would be beneficial for the authors to include a brief discussion on sample size limitations (14 residents, 11 family members, and 34 staff) when generalizing findings, even though qualitative studies typically value depth over breadth. Moreover, incorporating a summary at the end of the Results section (around lines 459–477) that links each theme to the research questions would clarify how the data supports the conclusions on acceptability and feasibility.

2. Appropriateness and Rigor of the Statistical (Descriptive) Analysis

The study mainly employs qualitative thematic analysis and uses descriptive statistics (frequencies and percentages, as seen in lines 191–198 and Table 1) to outline participant demographics. This approach is entirely suitable, considering that the study is not aimed at testing quantitative hypotheses.

Braun and Clarke’s six-step reflexive thematic analysis is well defined and seemingly followed rigorously. The iterative team coding with NVivo is clearly outlined (lines 181–190), reassuring the reader of the robustness of the thematic findings.

Descriptive statistics were employed to present demographics (lines 222–235), which is appropriate.

Given the qualitative nature of the study, advanced statistical analyses are not anticipated. However, the authors might briefly discuss how they ensured trustworthiness in the descriptive analysis (e.g., through triangulation, member checking, or inter-coder reliability). Such clarification would address potential concerns regarding methodological rigor. Integrating quantitative outcome measures (e.g., validated scales for mobility or affect), as suggested in the limitations section (lines 600–604), would also strengthen the study.

3. Data Availability and Transparency

The detailed descriptions provided in the Methods (lines 157–174) and supplemental materials reveal that raw interview protocols, field notes, and coding schemas are accessible, which is commendable for a qualitative study. While the paper discusses data rigor (lines 213–220) and ethical protocols (lines 205–212), it does not explicitly state whether transcripts, coding frameworks, or anonymized data are available in a public repository.

4. Clarity, Coherence, and Use of Standard English

The manuscript is well-structured and logically organized, featuring clear research questions (lines 92–96), a comprehensive methodology section, and results that flow logically.

The language throughout is accessible, standard, and grammatically correct, making it suitable for an international journal.

Complex concepts like feasibility and acceptability are well-defined and consistently applied (lines 78–91). The use of participant quotations enhances clarity and reader engagement (e.g., lines 243–249, 373–379). e.g., lines 243–249, 373–379).

Reviewer #6: This manuscript addresses a highly relevant issue in the context of institutionalized elder care by exploring the feasibility and acceptability of a gamified exercise intervention. The qualitative approach is appropriate, and the study demonstrates strong implementation, particularly through the inclusion of multiple stakeholder perspectives: residents, family members, and care staff. The inclusion of participants with dementia and the thoughtful adaptation of the technological tool to their needs are important strengths. Overall, the study is meritorious and provides original evidence. Nevertheless, I suggest minor revisions aimed at clarifying and strengthening some theoretical and analytical elements, as outlined below.

Theoretical integration: use of multiple conceptual frameworks

Throughout the manuscript, several frameworks are referenced: the Consolidated Framework for Implementation Research (CFIR), the Technology Acceptance Model (TAM), and the Unified Theory of Acceptance and Use of Technology (UTAUT). However, the relationship among these frameworks is not entirely clear. The manuscript states that the analysis was guided by CFIR while also indicating that themes were developed inductively. In addition, TAM and UTAUT appear later in the discussion as contextual frameworks. This coexistence may generate confusion regarding the overarching epistemological orientation of the study. I recommend clarifying how these frameworks were integrated in the design, analysis, and interpretation stages.

Distinction between acceptability and feasibility

While the authors argue that acceptability and feasibility are interconnected, the boundaries between the two are not always clearly defined in the manuscript. Specifically, the theme ease of use is repeatedly associated with both resident satisfaction (acceptability) and staff-related implementation concerns (feasibility). In such cases, it would be helpful to explicitly discuss how each concept is interpreted, the dimensions in which they manifest, and how they are empirically distinguished in the thematic analysis.

Generalizability and potential cultural bias in sampling

The sample appears to be demographically homogeneous, with all participants (residents, families, and staff) of Asian background, and the study was conducted in a single institution. Although the manuscript acknowledges limitations regarding generalizability, it does not explain why such a specific population was selected or what cultural or institutional implications this may have for interpreting the findings.

What was the rationale for selecting such a demographically homogeneous sample, particularly in terms of participants' cultural background? Was this decision deliberate, institutionally constrained, or a result of logistical limitations?

**Do you want your identity to be public for this peer review?** For information about this choice, including consent withdrawal, please see our Privacy Policy

Reviewer #5: **Yes: ** Adeniyi Adebayo

Reviewer #6: No

---

## [Author Response · Author response to Decision Letter 3]

10 Jun 2025

Reviewers' comments:

Reviewer #5: Hello author, thank you for providing this manuscript for review. I have shared several comments, primarily highlighting its strengths. However, I've also noted some areas where improvements could be made.

1. Technical Soundness and Data-to-Conclusion Alignment

The study is presented as a qualitative inquiry examining the acceptability and feasibility of a gamified cycling exercise program in a long-term care (LTC) setting. The authors carefully define essential concepts. For instance, in lines 78–91, "acceptability" is defined as how engaging and valuable the residents, family members, and staff find the intervention, whereas “feasibility" refers to practical aspects like staffing and scheduling. This dual focus is well justified in the context of the study.

The methodological approach is thoroughly described (e.g., an Interpretative Descriptive approach noted in lines 97–105 and the iterative coding process using NVivo discussed in lines 181–190). The methods for observations, individual interviews, and focus groups are suitably explained, and incorporating perspectives from multiple stakeholders (residents, families, staff, and leadership) enhances the conclusions drawn.

Quotations from participants (e.g., lines 245–254 and lines 378–391) effectively illustrate the emergent themes (such as ease of use, physical/mental health benefits, and enjoyment/community engagement) rooted in the data. Along with summary tables (Tables 1 and 2 referenced around lines 231 and 242), these data points provide credible support for the manuscript’s conclusions.

The conclusions regarding feasibility and acceptability (lines 638–650) are well grounded in the study findings and existing literature (e.g., lines 500–529).

Despite the overall strong technical design, it would be beneficial for the authors to include a brief discussion on sample size limitations (14 residents, 11 family members, and 34 staff) when generalizing findings, even though qualitative studies typically value depth over breadth. Moreover, incorporating a summary at the end of the Results section (around lines 459–477) that links each theme to the research questions would clarify how the data supports the conclusions on acceptability and feasibility.

Response: Thanks for your comments. We have added a brief reflection in the discussion addressing the sample size limitations (lines 622-627). We have also added a concise summary paragraph at the end of the results sections, explicitly linking each of the identified themes back to the research questions. We believe this addition improves the clarity and coherence of our findings (lines 468-494).

2. Appropriateness and Rigor of the Statistical (Descriptive) Analysis

The study mainly employs qualitative thematic analysis and uses descriptive statistics (frequencies and percentages, as seen in lines 191–198 and Table 1) to outline participant demographics. This approach is entirely suitable, considering that the study is not aimed at testing quantitative hypotheses.

Braun and Clarke’s six-step reflexive thematic analysis is well defined and seemingly followed rigorously. The iterative team coding with NVivo is clearly outlined (lines 181–190), reassuring the reader of the robustness of the thematic findings.

Descriptive statistics were employed to present demographics (lines 222–235), which is appropriate.

Given the qualitative nature of the study, advanced statistical analyses are not anticipated. However, the authors might briefly discuss how they ensured trustworthiness in the descriptive analysis (e.g., through triangulation, member checking, or inter-coder reliability). Such clarification would address potential concerns regarding methodological rigor. Integrating quantitative outcome measures (e.g., validated scales for mobility or affect), as suggested in the limitations section (lines 600–604), would also strengthen the study.

Response: Thanks for your comments. We have expanded the methods section to include a description of the strategies we used to ensure trustworthiness. (lines 190-192, 222-229)

3. Data Availability and Transparency

The detailed descriptions provided in the Methods (lines 157–174) and supplemental materials reveal that raw interview protocols, field notes, and coding schemas are accessible, which is commendable for a qualitative study. While the paper discusses data rigor (lines 213–220) and ethical protocols (lines 205–212), it does not explicitly state whether transcripts, coding frameworks, or anonymized data are available in a public repository.

Response: Thanks for your comments. We have added a clear statement to the ethical consideration section to clarify our approach. (lines 215-217). While full transcripts and anonymized datasets are not publicly available, we have included coding frameworks and selected de-identified excerpts in the manuscript to support transparency.

4. Clarity, Coherence, and Use of Standard English

The manuscript is well-structured and logically organized, featuring clear research questions (lines 92–96), a comprehensive methodology section, and results that flow logically.

The language throughout is accessible, standard, and grammatically correct, making it suitable for an international journal.

Complex concepts like feasibility and acceptability are well-defined and consistently applied (lines 78–91). The use of participant quotations enhances clarity and reader engagement (e.g., lines 243–249, 373–379). e.g., lines 243–249, 373–379).

Response: Thanks for your comments.

Reviewer #6: This manuscript addresses a highly relevant issue in the context of institutionalized elder care by exploring the feasibility and acceptability of a gamified exercise intervention. The qualitative approach is appropriate, and the study demonstrates strong implementation, particularly through the inclusion of multiple stakeholder perspectives: residents, family members, and care staff. The inclusion of participants with dementia and the thoughtful adaptation of the technological tool to their needs are important strengths. Overall, the study is meritorious and provides original evidence. Nevertheless, I suggest minor revisions aimed at clarifying and strengthening some theoretical and analytical elements, as outlined below.

Theoretical integration: use of multiple conceptual frameworks

Throughout the manuscript, several frameworks are referenced: the Consolidated Framework for Implementation Research (CFIR), the Technology Acceptance Model (TAM), and the Unified Theory of Acceptance and Use of Technology (UTAUT). However, the relationship among these frameworks is not entirely clear. The manuscript states that the analysis was guided by CFIR while also indicating that themes were developed inductively. In addition, TAM and UTAUT appear later in the discussion as contextual frameworks. This coexistence may generate confusion regarding the overarching epistemological orientation of the study. I recommend clarifying how these frameworks were integrated in the design, analysis, and interpretation stages.

Response: Thanks for your comments. Due to our focus on the implementation process of the gamified exercise intervention, we primarily used the CFIR to guide our thematic analysis. CFIR provided a structured lens to explore contextual, organizational, and individual factors influencing feasibility and acceptability within LTC settings. While other studies have employed TAM and UTAUT to examine technology adoption, these frameworks are not used in our data analysis. Instead, we referenced TAM and UTAUT in the discussion section as comparative models to assess whether our CIFR-informed themes aligned with constructs from these established technology acceptance frameworks. This comparison enabled us to interpret our findings within a broader body of literature, while maintaining an implementation-focused epistemological orientation. We made relevance revision to the discussion section (lines 547-582).

Distinction between acceptability and feasibility

While the authors argue that acceptability and feasibility are interconnected, the boundaries between the two are not always clearly defined in the manuscript. Specifically, the theme ease of use is repeatedly associated with both resident satisfaction (acceptability) and staff-related implementation concerns (feasibility). In such cases, it would be helpful to explicitly discuss how each concept is interpreted, the dimensions in which they manifest, and how they are empirically distinguished in the thematic analysis.

Response: Thanks for your comments. We have clarified how acceptability and feasibility are interpreted and distinguished in our thematic analysis. An explanation has been added to the end of the findings section to address this distinction (lines 486-491).

Generalizability and potential cultural bias in sampling

The sample appears to be demographically homogeneous, with all participants (residents, families, and staff) of Asian background, and the study was conducted in a single institution. Although the manuscript acknowledges limitations regarding generalizability, it does not explain why such a specific population was selected or what cultural or institutional implications this may have for interpreting the findings.

What was the rationale for selecting such a demographically homogeneous sample, particularly in terms of participants' cultural background? Was this decision deliberate, institutionally constrained, or a result of logistical limitations?

Response: Thanks for your comments. The demographic homogeneity of our sample was not an intentional selection but a natural consequence of recruiting at the available site. The LTC facility we partnered with serves a predominantly Chinese-Canadian community, which naturally shaped the participant composition. While this reflects the local institutional and cultural context, it was not a deliberate choice to limit diversity. We recognize that this homogeneity may influence the generalizability of our findings and have emphasized the importance of future research involving more culturally diverse populations and multiple sites to explore broader applicability and cultural nuances in acceptability and feasibility. We added this explanation to the limitation section (lines 619-622).

---

## [Decision Letter · Decision Letter 3]

17 Jul 2025

Dear Dr. Hung,

Thank you for submitting your manuscript to PLOS ONE. After careful consideration, we feel that it has merit but does not fully meet PLOS ONE’s publication criteria as it currently stands. Therefore, we invite you to submit a revised version of the manuscript that addresses the points raised during the review process.

We look forward to receiving your revised manuscript.

Kind regards,

Amir Karimi, PhD

Academic Editor

PLOS ONE

Journal Requirements:

**Additional Editor Comments:**

Hello dear authors, please correct and submit the reviewer's comments for final acceptance.

Reviewers' comments:

Reviewer's Responses to Questions

**Comments to the Author**

Reviewer #7: All comments have been addressed

2. Is the manuscript technically sound, and do the data support the conclusions?

Reviewer #7: Yes

3. Has the statistical analysis been performed appropriately and rigorously?

Reviewer #7: Yes

4. Have the authors made all data underlying the findings in their manuscript fully available?

Reviewer #7: Yes

5. Is the manuscript presented in an intelligible fashion and written in standard English?

Reviewer #7: Yes

Reviewer #7: The authors' research is of great help in improving the welfare of the elderly. Several reviewers made many useful comments, and the authors responded accordingly. In a word, the study was well done. I suggest that in future studies, the sample and scope of the study should be enlarged, and more cutting-edge statistical methods should be utilized to conduct in-depth studies to explore the mechanism.

**Do you want your identity to be public for this peer review?** For information about this choice, including consent withdrawal, please see our Privacy Policy

Reviewer #7: No

---

## [Author Response · Author response to Decision Letter 4]

10 Oct 2025

Dear Editorial Team,

Thank you very much for your time and support in handling our manuscript titled “Feasibility and acceptability of gamified cycling exercise for residents in a long-term care home: a qualitative study” (Manuscript ID: PONE-D-24-48645R3).

We have carefully reviewed the contents of the manuscript and uploaded the latest version, ensuring that it contains no tracked changes or highlighting.

Thank you again for your kind consideration and for the constructive feedback that has helped strengthen our work.

Sincerely,

Lillian

IDEA lab, UBC

---

## [Editor Report · Decision Letter 4]

15 Oct 2025

Feasibility and acceptability of gamified cycling exercise for residents in a long-term care home: a qualitative study

PONE-D-24-48645R4

Dear Dr. Hung,

We’re pleased to inform you that your manuscript has been judged scientifically suitable for publication and will be formally accepted for publication once it meets all outstanding technical requirements.

Kind regards,

Amir Karimi, PhD

Academic Editor

PLOS ONE
---

## [Editor Report · Acceptance letter]

PONE-D-24-48645R4

PLOS ONE

Dear Dr. Hung,

I'm pleased to inform you that your manuscript has been deemed suitable for publication in PLOS ONE. Congratulations! Your manuscript is now being handed over to our production team.

Kind regards,

on behalf of

Dr. Amir Karimi

Academic Editor

PLOS ONE